# Acceptability and playability of an organization training videogame for young adolescents with ADHD: The development of ATHEMOS

**Brandon K. Schultz**[1]*, **Steven W. Evans**[2], **John Bowditch**[3], **Kaitlynn Carter**[1], **Emma E. Rogers**[2], **Jennifer Donelan**[1], **Allison Dembowski**[1]

**1** Department of Psychology, East Carolina University, Greenville, North Carolina, United States of America, **2** Department of Psychology, Ohio University, Athens, Ohio, United States of America, **3** Game Research and Immersive Design Laboratory, Ohio University, Athens, Ohio, United States of America

* SchultzB@ecu.edu

## Abstract

An estimated 8.7% to 9.8% of school-age children in the United States have attention deficit hyperactivity disorder (ADHD), affecting 4.3 to 4.9 million public school students. ADHD is a costly disorder that often goes untreated, especially among adolescents. Accessible computer-based programs have emerged to address the neurocognitive deficits of ADHD, but results to date have been disappointing. In this study, we tested the acceptability, playability, and user satisfaction of a novel planning/organization skills training game, called "ATHEMOS," based on an established psychosocial treatment package (i.e., *Challenging Horizons Program*). We conducted eight focus groups during a three-year development period, using feedback from 72 young adolescents with ADHD to iteratively improve the game. Then, during a pilot study in the fourth year, we collected data from 16 young adolescents who played the game as part of a 6- to 16-week school-based intervention. Our findings suggest that the serious game resulted in acceptability and playability ratings only moderately below that of recreational games ($\delta = -0.40$). Critically, average perceptions remained positive when delivered within a school-based ADHD intervention over several weeks or months, with strong overall user satisfaction. Boys found the game more acceptable than girls, with ratings near that of recreational games ($\delta = -0.23$). We conclude that computer-assisted behavior interventions appeal to adolescents with ADHD and offer a potentially promising treatment delivery method in schools.

**Data Availability Statement:** Data are available at https://osf.io/ew7qk/.

## Author summary

Attention-Deficit Hyperactivity Disorder (ADHD) causes academic impairments, and yet often goes untreated. The *Challenging Horizons Program* (CHP) was developed to address this need by teaching specific coping skills (e.g., organization, planning, study strategies) to reduce the impairing aspects of ADHD. Although effective, the CHP is too costly for

**Funding:** The research described in this article was supported by a grant (R324A180219) from the US Department of Education Institute of Education Sciences (IES) to East Carolina University ($773,968 to BKS) and Ohio University ($615,456 to SWE). IES had no role in the research methodology, data collection or analysis, or preparation of this manuscript.

**Competing interests:** The first two authors, BKS and SWE, are exploring options to potentially redesign and commercialize the videogame and intervention package described herein but have no active commercial interests at the time of submission.

many school districts. In this development project, we examined the acceptability and playability of a videogame-based alternative that trains the organization and planning skills of the CHP in a playful manner. Over a three-year development period we created the game, called ATHEMOS, using feedback from 72 young adolescents with ADHD. Then, in a small pilot study, we deployed ATHEMOS as part of a teacher-led intervention in middle schools. The results suggest that young adolescents with ADHD enjoy ATHE-MOS almost as much as leisure games and were satisfied with the overall intervention package. We conclude that serious games teaching organization and planning skills are acceptable and might offer a promising new direction for affordable school-based treatment.

## Introduction

An estimated 8.7% to 9.8% of school-age children in the United States have attention deficit hyperactivity disorder (ADHD), affecting 4.3 to 4.9 million public school students [1]. ADHD is characterized by developmentally inappropriate levels of inattention and/or hyperactivity/ impulsivity, beginning prior to age 12, and leading to functional impairments at school and home [2]. Research shows that ADHD is a chronic disorder, with 70% to 87% of children with ADHD continuing to exhibit symptoms and impairments in adolescence [3] and a substantial proportion developing comorbid conduct problems, especially among boys [4]. School-based behavior interventions can alleviate some of the academic and social impairments of ADHD, but there are persistent implementation challenges that require interventions to be designed and tested specifically in those settings [5].

The *Challenging Horizons Program* (CHP) is an evidence-based treatment for ADHD designed for use in middle schools. The training interventions (TI) that comprise the CHP— organization, assignment tracking, note taking, study skills, and social skills—were initially developed and tested in a 4.5-hour per week after-school program [6]. The program is staffed by trained paraprofessionals who deliver TI in accordance with a treatment manual, typically under the supervision of a clinical or school psychologist. Studies show that the after-school version of the CHP is effective, resulting in meaningful improvements in organization, home-work completion, academic progress, and attention-related behavior when compared to typical school-based practices ($d$s = .24 to .63) [7] Participants who comply with the CHP— defined as 80%+ attendance during 1-year of intervention—show especially large gains when compared to statistically matched students in a treatment-as-usual control group ($d$s = 0.56 to 2.00) [8].

Although effective, staffing costs for the CHP after-school program are prohibitive for most schools [9]. For this reason, a cost-effective consultation version of the program (CHP-C) was developed for use during the school day. In the CHP-C, trained behavioral consultants support teachers as they implement CHP interventions in their classrooms, meeting with those teach-ers regularly to discuss the interventions, track intervention data, and troubleshoot implemen-tation challenges. The CHP-C has been shown to be efficacious in a prospective randomized control trial of middle school students with ADHD, with cumulative benefits over time when compared to typical practices [10]. For example, parent ratings of inattention, hyperactivity, and social functioning suggested small to large benefits ($d$s = 0.24 to 1.05) for CHP-C partici-pants. Students with ADHD who did not receive the CHP-C were roughly four times more likely than the treatment group to have their grade point averages drop below the threshold for failing in each grading period [11]. But a tradeoff of the CHP-C is that participants receive

relatively low doses of intervention, and some benefits (e.g., ADHD symptom reduction) can take up to three years to materialize [10].

There are several reasons why the CHP-C delivers a lower dose than the CHP. First, most teachers cannot devote several hours each week to one-to-one meetings with individual students. Second, many teacher consultees report that participants either skip meetings or fail to bring the requisite materials (e.g., bookbags, binders). So, although dropouts are rare (below 3%), participants may not engage well with teachers [12]. Third, teachers find some of the procedures aversive because student performance is typically poor at first, requiring the teacher to repeatedly correct errors, which can strain the student-teacher relationship [13]. Fourth, like all forms of school consultation, the CHP-C is susceptible to poor implementation fidelity. For example, teacher consultees are often slow to start interventions, in some cases for months, because they are unfamiliar with TIs for ADHD. Early CHP-C consultation sessions are often spent coaching and reassuring teacher consultees [7]. Thus, we explored options to deliver the CHP in a manner that is less burdensome for teachers and acceptable to adolescents with ADHD.

Computer-based programs have emerged to support and strengthen school-based intervention, but most options for ADHD to date target the neurocognitive correlates of the disorder (e.g., working memory deficits), with disappointing results [14]. Designers have assumed that gains in neurocognitive functioning can alleviate ADHD impairments in real-world settings, but the research to date has fallen short of expectations [15, 16]. As an alternative, serious games might deliver TIs that directly remediate functional impairments, and then those new skills might be transferred to a target setting with the active assistance of interventionists (e.g., teachers, coaches). However, we are aware of only one game in the research literature that delivers behavioral skill-training interventions, and it is only available for Dutch-speaking players [17, 18].

Our goal was to develop an intervention package consisting of an engaging videogame and real-world intervention, which we refer to as computer-assisted behavior intervention (CABI). The game, called "ATHEMOS," is an effort to train players in the planning, organization, and note-taking skills taught in the CHP, but ideally introduced in an engaging and playful manner. After players are exposed to the concepts in the game, teacher mentors help players transfer these skills to the classroom. The work of the teacher mentors is outlined in a treatment manual that describes the game and provides strategies for helping players apply the skills in the real world. To our knowledge, ATHEMOS is the first serious game in English that trains organization and planning skills for students with ADHD, paired with school-based intervention to achieve skill transfer.

Given the novelty of the CABI concept, it was not clear if adolescents with ADHD would find a gamified training intervention engaging, or whether interest in the game might wane once delivered within a school-based intervention package. There is no advantage to gamifying an intervention like the CHP if the product is unacceptable to the target population. Thus, our focus in this development study was the degree to which ATHEMOS and its broader school-based intervention package appeals to young adolescents with ADHD. Our specific aims were to (a) compare focus group results to the average total score values of a validated videogame satisfaction scale to assess the degree to which the playability and acceptability of ATHEMOS approximates that of recreational videogames; and (b) examine player perceptions of the game when implemented within a school-based intervention package to determine if playability, acceptability, and satisfaction with the game are negatively impacted. The second aim was achieved in a small, randomized pilot study (described below), and for our purposes here we will focus on the pilot study participants who received the game. We anticipated that boys and girls might have differing impressions of an impairment-focused intervention, given the

differing experiences of boys and girls with ADHD, but also given that boys are generally more likely than girls to report videogame interest and use [19, 20].

## Method

All study procedures were pre-approved by the University and Medical Center Institutional Review Board at East Carolina University (UMCIRB #18-001555) and the Institutional Review Board at Ohio University (IRB #21-F-28).

### Study design

Seventy-two young adolescents with ADHD, ages 10-15 years, were recruited to participate in focus groups during the early stages of game design and playtesting. Beginning early in the fall of 2019, focus groups were advertised to all families of middle school-age patients with ADHD at an urban, mid-Atlantic pediatric outpatient clinic that accepts public insurance. A nurse at the clinic reviewed medical records to identify patients with ADHD diagnoses and then contacted those families directly to announce the study. In addition, clinic staff ensured that study announcements (i.e., flyers) were available in the waiting rooms. Interested caregivers called a dedicated research line and were screened by a project coordinator to ensure that adolescents exhibited four or more of the criteria for inattentive symptoms, as assessed using the relevant section of the *Disruptive Behavior Disorders Rating Scale* [21]. Families were also asked to confirm their child's age, diagnosis, and English-speaking status before being scheduled to participate.

On the day of the focus group, families provided written parent consent and child assent, and then participants completed the *Revised Computer Game Attitude Scale* (RCGAS) [22] to assess videogame confidence and experience. The RCGAS comprises 17 items using 5-point Likert-type response formats, ranging from *strongly disagree* (1) to *strongly agree* (5), with factors assessing videogame confidence, perceptions of educational videogames, perceived enjoyment from videogames, and perceptions of videogames for leisure. After the RCGAS was completed, participants playtested ATHEMOS while parents completed the *ADHD Rating Scale-5th Edition* (ADHD-5) [23] to assess ADHD symptom severity. Most families completed the focus group within 1-hr and all were paid $75 to reimburse their time and travel.

In the final year of the project, a school-based pilot study (*n* = 16) was conducted to assess user reactions to the game when used within the context of a school-based intervention. Researchers recruited this group of participants from two middle schools, one in North Carolina, USA, and one in Ohio, USA, producing a unique sample. Study announcements were distributed to families through multiple means, including automated calls from principals, social media, and flyers. Interested families contacted our research labs, completed a screening call identical to the one used for the focus groups, and then scheduled an intake assessment to confirm the ADHD diagnosis. During the intake, families completed written consent/assent and then were administered the RCGAS and the ADHD-5. Participating families received $75 for this initial intake assessment. All demographic data were collected from parents using a standard form.

No significant differences were noted between the focus group and pilot study participants on any of the intake variables except for racial composition; specifically, focus group participants were more likely to be students of color (76.4%) than the pilot study participants (50.0%). We examined responses to the ADHD-5 or RCGAS and found no race-related differences, so we believe the varying racial compositions across our groups was inconsequential. A large proportion (38.9%) of parents in the focus groups did not report their family income, so we examined mother's level of education as a proxy for socioeconomic status and found no

**Table 1. Demographics, ADHD Severity, and Computer Game Attitudes of the Focus Group and Pilot Study Participants (*n* = 88).**

| | Focus Groups (*n* = 72) | | Pilot Study (*n* = 16) | |
|---|---|---|---|---|
| | *n* | % | *n* | % |
| Sex | | | | |
| Boys | 52 | 72.2 | 10 | 62.5 |
| Girls | 20 | 27.8 | 6 | 37.5 |
| Race | | | | |
| Black | 52 | 72.2 | 7 | 43.8 |
| White | 17 | 23.6 | 8 | 50.0 |
| Biracial | 2 | 2.8 | 0 | 0 |
| Alaskan Native/Native American | 1 | 1.4 | 1 | 6.3 |
| Ethnicity | | | | |
| Hispanic | 4 | 5.6 | 0 | 0 |
| Non-Hispanic | 59 | 81.9 | 14 | 87.5 |
| No response | 9 | 12.5 | 2 | 12.5 |
| Mother's Education | | | | |
| Graduate professional training | 6 | 8.3 | 0 | 0 |
| 4-year college or university | 11 | 15.3 | 2 | 12.5 |
| 1-3 years college/university | 33 | 45.8 | 9 | 56.3 |
| High school graduate | 17 | 23.6 | 4 | 25.0 |
| Some high school | 5 | 6.9 | 1 | 6.3 |
| | *M* | *SD* | *M* | *SD* |
| Age | 12.9 | 1.2 | 12.7 | 1.0 |
| ADHD Rating Scale-5 | | | | |
| Inattention | 18.6 | 7.1 | 20.8 | 4.6 |
| Hyperactivity-Impulsivity | 13.6 | 6.9 | 15.4 | 7.0 |
| Revised Computer Game Attitude Scale | | | | |
| Confidence | 3.9 | 0.8 | 4.0 | 1.1 |
| Learning | 4.0 | 0.9 | 4.1 | 0.7 |
| Liking | 3.9 | 0.9 | 4.1 | 0.6 |
| Leisure | 3.6 | 1.0 | 4.0 | 0.9 |

differences across groups. Demographic data and intake rating scales results for the focus group and pilot study participants are summarized in Table 1.

## Procedures

**Game development.** ATHEMOS was conceived by the first author (BKS) in collaboration with the second author (SWE) and created by student game designers at the Game Research and Immersive Design (GRID) Lab at Ohio University. In total, 22 students majoring in game development, animation, and music production were employed by the GRID Lab at various points in the project. Game development included designing story and play, programming, art creation, sound design, and quality testing. Progress was slowed by the COVID-19 pandemic, stretching the production time frame from 4 months for design and 8 months for programming and asset creation to over 18 months total. Testing began at the start of production and continued several months past the end of principal production. The design team worked with a continually changing design document to manage the project's scope, with themes, characters, locations, and puzzles primarily designed by the first author.

The Unity real-time development platform was used to create ATHEMOS, and the code was written in C# using Microsoft Visual Studio and GitHub Enterprise. Digital art was created with Autodesk Maya and Adobe Photoshop, while most animations were created using the GRID Lab's 24-camera OptiTrack motion capture system and Autodesk MotionBuilder and Motive software. Sound recording and mixing were done using Avid Pro Tools and Yamaha Nuendo. Bugs, typos, and inconsistencies found during testing were logged on a spreadsheet and assigned to team members to fix. Project management and team communication were managed using Microsoft Teams and HacknPlan. The final product was designed for the Windows 10 operating system, on a PC with a discrete graphics card and 8GB RAM or higher, using a two-joystick gamepad.

**Game narrative.**   We chose a game narrative we anticipated would appeal to boys because boys with ADHD outnumber girls with ADHD at a ratio of approximately 2:1 [2]. Problems of disorganization also tend to be more severe and more common among boys than girls [4]. In the game, mysterious extraterrestrials threaten Earth from various locations around our solar system. The player pilots a new type of spacecraft, called the "ATHEMOS," to collect information (i.e., "intel") about the extraterrestrials. To gather intel, the player must disable alien robot drones during space battles (Fig 1). The space battle element of ATHEMOS is recreational and intended to motivate the player, with difficulty increasing each round and readjusting when players struggle. Afterward, the player returns to Earth and works with the "Silent Canopy" taskforce, an international team of scientists, linguists, engineers, and cryptologists, to organize and interpret the intel (Fig 2). This cycle (i.e., "day") is repeated with the player engaging in space battles and collecting intel until all pieces of information are correctly organized and the player learns why the aliens are attacking.

**Minigame 1: Scheduling.**   Each game cycle, the player meets with one of four non-player characters (NPCs) who assist in understanding the intel. Players are required to keep a weekly schedule of appointments with the NPCs for these special "debrief" meetings. The player chooses their schedule, but once set, the player must meet with the taskforce NPC assigned that day. Each "Monday" players are required to reset their weekly schedule. During this

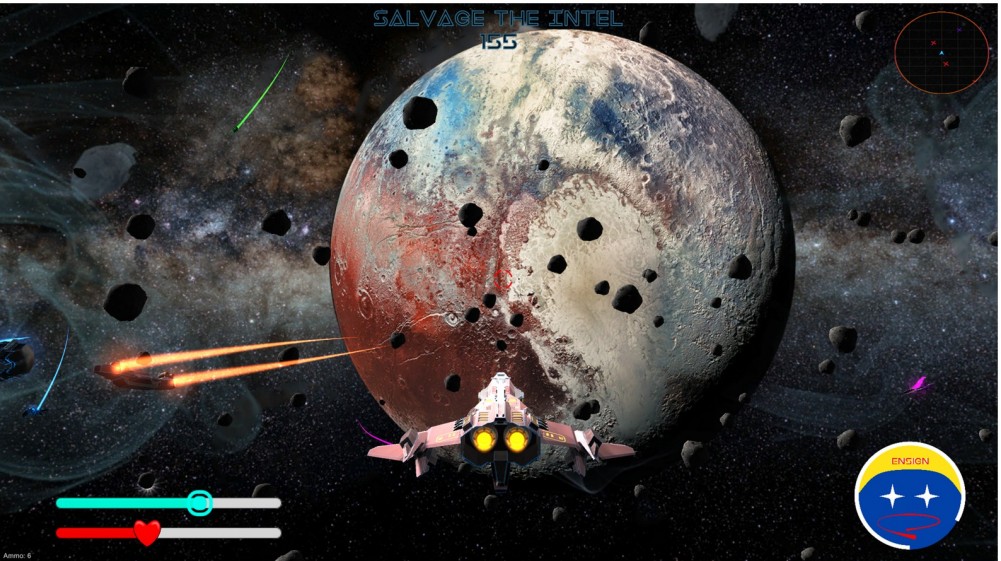

**Fig 1. Screenshot of the Space Battle Sequence in ATHEMOS.**

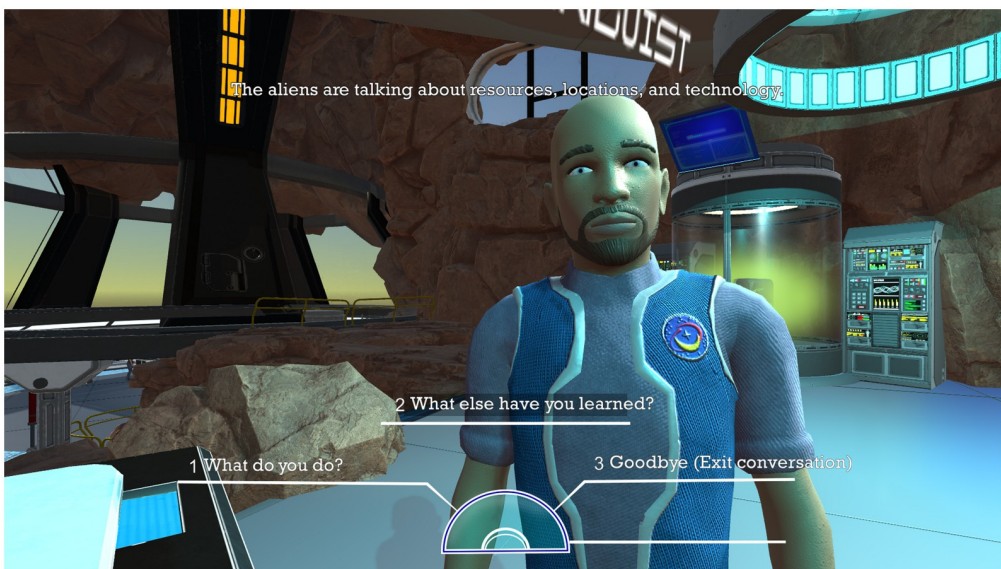

**Fig 2. Screenshot of Player Interaction with the NPCs in ATHEMOS.**

minigame, players place NPC tiles into a grid analogous to a weekly calendar (Fig 3). Although there are no correct or incorrect responses, the choice of meetings can either expedite or slow the player's progress in the note taking minigame (see below), introducing a strategic element in the game to ideally strengthen players' problem-solving skills [24] The scheduling minigame is matched to a real-world assignment tracking intervention intended to help players track schoolwork needs. Young adolescents with ADHD often have little prior experience with setting schedules independently, and often misunderstand how to arrange a planner, or the importance of checking it regularly. The purpose of this minigame is to rehearse a simple

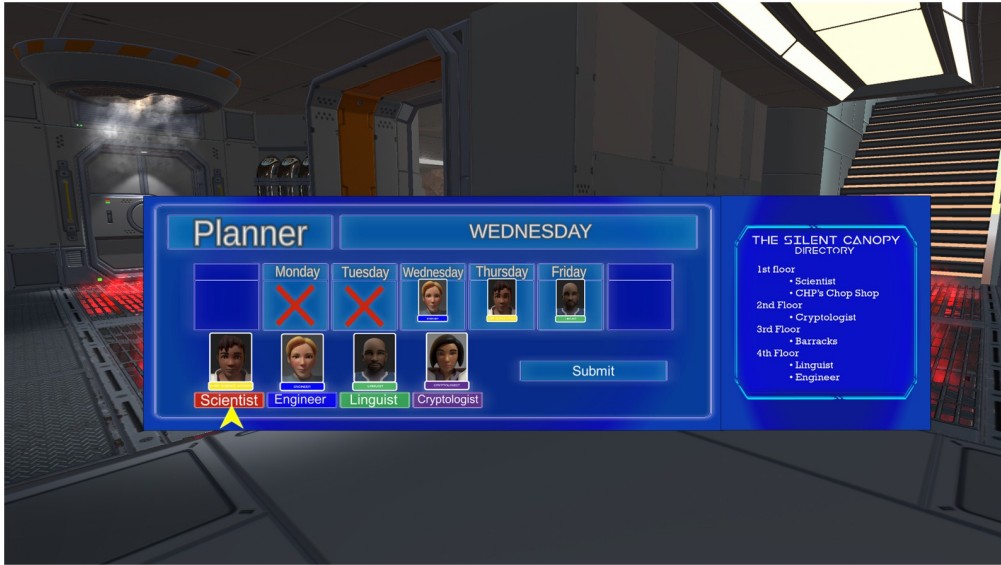

**Fig 3. Screenshot of Educational Element in ATHEMOS showing the Planning Minigame.**

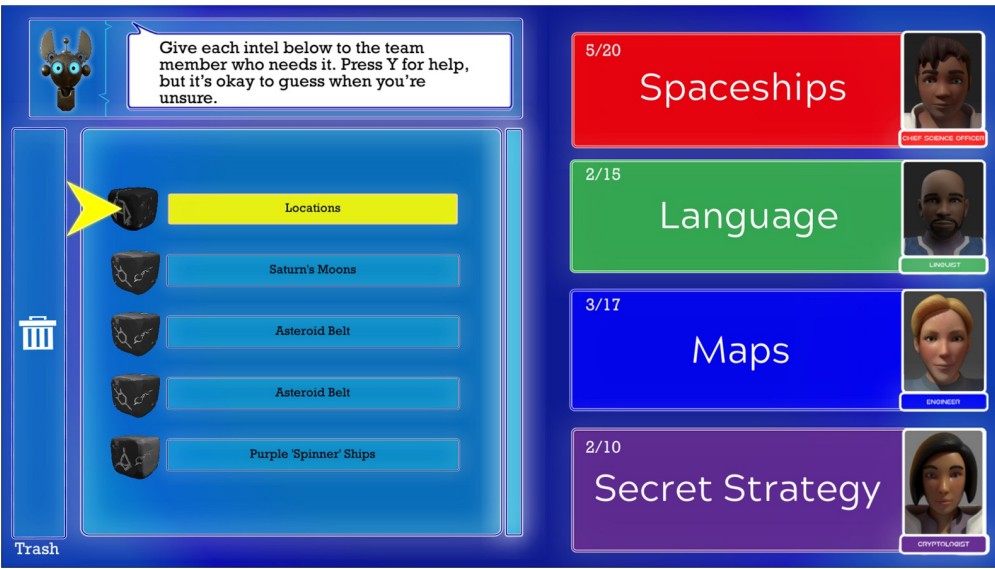

**Fig 4. Screenshot of Educational Element in ATHEMOS showing the Organization Minigame.**

weekly planner, thereby providing an example for the interventionist to use when introducing and tracking success in a real-world school planner.

**Minigame 2: Organization.** The game "intel" are discrete noun and adjective phrases that, when sorted correctly, provide an outline of the aliens' backstory. Each intel fits into one of four categories–spaceships, language, maps, and secret strategy–but not always in an obvious manner; players must take advantage of clues provided in the Silent Canopy base to sort everything correctly. In the organization minigame, players sort the intel into the four categories (Fig 4). Players are allowed to guess, with any mistakes corrected in later cycles. The four categories are analogous to the four core courses in school: Science, Language Arts, Math, and Social Studies. The colors associated with those categories–red, green, blue, and purple–are ideally matched to notebooks in a real-world binder system that the interventionist helps the player develop and maintain. To further help the transfer of learning, the interventionist can use ATHEMOS-themed checklists and stickers in the student's real-world binder. Young adolescents often struggle with organization, particularly multi-subject binders. The purpose of this minigame is to introduce a color-coded, four-subject organization system and provide the player with opportunities to rehearse categorical sorting.

**Minigame 3: Note taking.** During the scheduled debrief meetings with NPCs, players further sort intel into main ideas and supporting details. The note taking minigame requires players to place the intel on an outline-style grid, with main ideas next to Roman numerals and supporting details next to letters, like a classic outline (Fig 5). Again, the player is allowed to guess, with all mistakes in this minigame removed and returned to the organization minigame (described above). While the player works, an audio voiceover reveals the backstory relevant to this category and functions much like a classroom lecture as the student records notes. Progress in this minigame is rewarded with in-game upgrades, like new designs for the ATHEMOS. Overall game progress is linked to performance on this minigame, and the game ends once players place all main ideas and supporting details in the correct spaces in all four categories. Young adolescents with ADHD often struggle with classroom note taking, which can lead to disruptive behavior. The goal of this minigame is to introduce a note taking format and

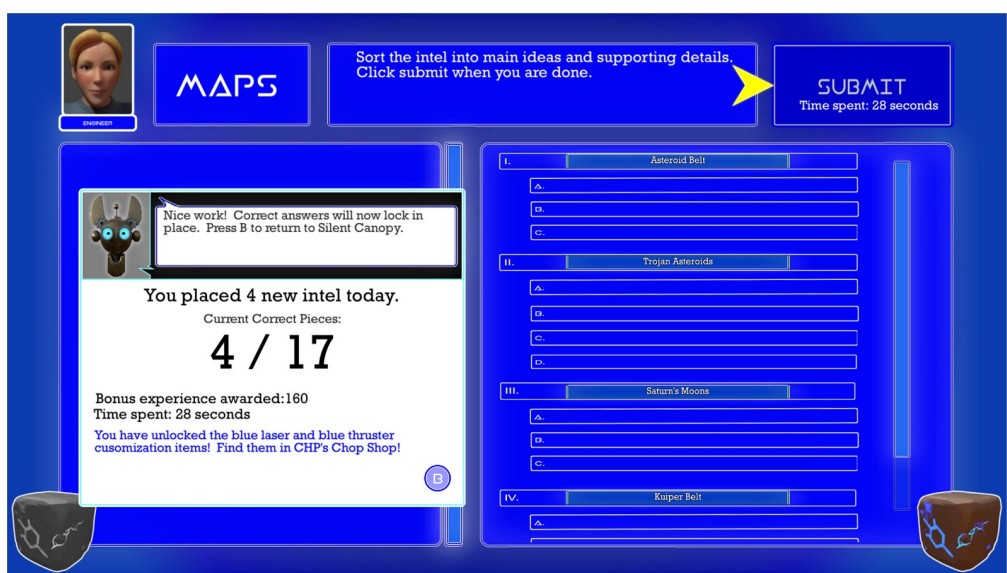

**Fig 5. Screenshot of Educational Element in ATHEMOS showing the Note taking Minigame.**

allow the player to rehearse outlining information presented didactically. The interventionist can use this experience as an example for how players can track materials taught in their classrooms, and then reward real-world success accordingly.

**Focus groups.** After two in-person focus groups in the fall of 2019/winter of 2020, conducted in a classroom on a university campus, social distancing and other public health measures were instituted in response to the COVID-19 pandemic. Once we were able to resume, we altered our procedures and invited families to participate one at a time. Plans to audio record and code responses to open-ended questions were canceled and replaced by questionnaires. To make the data comparable over time, we recruited roughly 10 participants to play the same version of the game before switching to an updated version of the game, on a rolling basis. Once feedback from ≈ 10 participants was collected (range = 5 to 14), the data were shared with the design team, the game was revised, and a new version was created for the next set of focus group participants. Mostly these iterative changes focused on improvements to the user interface (e.g., puzzle controls), in-game tutorials, and conveying the game narrative effectively.

The design team also analyzed event-based user logs instrumented in ATHEMOS that recorded several aspects of in-game user behavior. User logs captured performance metrics related to game stability (e.g., frame rate) and potential bugs (e.g., crash events, warnings, stuck events) [25]. Using these data, we assessed the stability of the game and revised as needed. User logs also allowed the design team to assess game difficulty. We originally planned the game for an 8- to 10-hour user experience, and adjusted elements to approximate this target runtime, including the amount of intelligence available during each cycle, the difficulty of the puzzles, and the help provided by the NPCs. Main elements of the game (e.g., narrative, locations, NPCs) were completed prior to the first focus group and remained largely unchanged.

**Pilot study.** After a three-year development period, we tested the game within the context of a school-based intervention package. Each participant in this part of the study was matched with a teacher "mentor" who oversaw the intervention and helped the participant transfer

skills learned in the game to the real world. Participant-teacher pairings were guided by participant preferences, but also teacher availability and willingness. Recruitment for the pilot study began in the fall of 2021, but intervention did not begin for most participants until January 2022. Recruitment continued on a rolling basis, with some participants not starting intervention until April 2022. Four trained doctoral students (two at each site) provided ongoing school consultation during the intervention period, consistent with CHP-C procedures, meeting with teachers on a weekly to biweekly basis to track intervention data and problem-solve implementation challenges. All computer equipment, including laptops and game controllers, was provided to the schools by the research team.

## Dependent measures

**Game User Experience Satisfaction Scale (GUESS).** The *Game User Experience Satisfaction Scale* (GUESS) [26] is a 55-item scale used for game evaluation purposes. It comprises nine subscales: Usability/Playability, Narratives, Play Engrossment, Enjoyment, Creative Freedom, Audio Aesthetics, Personal Gratification, Social Connectivity, and Visual Aesthetics. Items use seven-point Likert-type response formats, ranging from *Strongly Disagree* (1) to *Strongly Agree* (7), with an additional not-applicable option. High scores are associated with game satisfaction (with one item reverse-scored). The GUESS was validated with 1,400 adult gamers who rated over 450 recreational videogames across popular genres (e.g., role-playing, action-adventure). Respondents rated a videogame they had voluntarily played for at least 10 hours, which presumably indicated enjoyment and commitment. The GUESS was found to have high content validity and good internal validity, as evidenced by an exploratory factor analysis and follow-up confirmatory factor analysis [26]. Moreover, a composite total score, derived from all the GUESS items, has been used to measure satisfaction during gameplay and has shown excellent reliability [27].

In the present study, we did not administer the Play Engrossment and Social Connectivity subscales of the GUESS. In the first instance (Play Engrossment), the items measure the degree to which the player is captivated by the game, but some items presume the player has unlimited time to play, which was not true in either our focus groups or pilot study. In the second instance (Social Connectivity), the items measure the degree to which the game encourages social connection, but that is not a goal of ATHEMOS. As a result of these adjustments, we will refer to the domains measured by the GUESS as *playability* and *acceptability*, based on the retained subscales. We also recalculated the total score mean of Phan and colleagues' [26] factor analysis results (hereafter *reference data*) using a weighted average of the items in the seven subscales we administered, which we estimated at 5.71. Internal reliabilities for the seven subscales and the total score in our data were acceptable to good ($\alpha$ = .75 to .84), with only one subscale (Audio Aesthetics) falling below the .80 threshold.

**Satisfaction questionnaire.** Satisfaction ratings were collected from the 16 pilot study participants using a brief, 8-item questionnaire previously developed by the authors for the CHP-C. Items on the questionnaire were modified slightly in the present study to reference ATHEMOS (e.g., "How satisfied are you with the amount of help you received through playing the game and working with the mentor?" "If a friend were in need of similar help, would you recommend our intervention and game to them?"). Each item uses a 4-point Likert response format, with higher ratings indicating greater satisfaction. The higher scores are anchored with statements such as *very satisfied* or *yes, definitely*, whereas lower scores are anchored with statements such as *poor*, or *quite dissatisfied*. The internal reliability of this instrument in the current study was good ($\alpha$ = .86).

## Statistical analyses

To start, participants' demographic data were summarized with descriptive statistics. Then, to determine the degree to which the current sample approximated the reference data on the GUESS, we applied Bayesian one-sample t-tests and updated our estimates with each focus group and then the pilot study. To compare the focus and pilot study groups, we applied Bayesian ANCOVA, controlling for participant sex. Unlike frequentist statistics, Bayesian tests can estimate the strength of evidence for both the alternative and null hypotheses using Bayes Factors (BF). BF can be interpreted like odds ratios, with values above 1.0 suggesting that the evidence favors the alternative hypothesis (i.e., group dissimilarity), with values exceeding 10.0 suggesting strong evidence for dissimilarity. Likewise, values below 1.0 suggest that the evidence supports the null hypothesis (i.e., group similarity), with values below .10 suggesting strong evidence for similarity. When the null and alternative hypotheses are equally supported by the data, the BF is 1.0. The magnitude of the effect is estimated with a standardized effect size ($\delta$), which is the population equivalent of Cohen's $d$ [28]. Data screening and some tests of parametric assumptions were conducted in SPSS (v.28) [29], and all Bayesian statistics were calculated using JASP (v. 0.16.4) [30].

# Results

We examined results from our eight focus groups as those data were collected. There were no clear patterns of increasing or decreasing satisfaction with ATHEMOS across the eight focus groups, despite iterative game revisions during that same period. Participants were generally satisfied with ATHEMOS as measured on the 7-point GUESS total score scale (total $M = 5.28$; $SD = 1.07$) beginning with the first focus group. Overall, 68% of focus group respondents' total scores were 5.0 or higher, and 24% of respondents' total scores were 6.0 or higher. The highest user ratings were found on the Enjoyment subscale ($M = 5.52$; $SD = 1.24$), and the lowest ratings were found on the Narratives subscale ($M = 5.10$; $SD = 1.17$).

To assess how strongly the total score results approximate recreational games, we compared our focus group data to the GUESS reference data average [26]. We noted that our focus group data were negatively skewed, as evidenced by boxplots and the Kolmogorov-Smirnov test ($D$ [72] = .132, $p$ = .003), so we applied a one-directional Bayesian Wilcoxon signed-rank test, anticipating the game would perform below the recreational games standard. Because we were unsure how far below the standard ATHEMOS would perform, we applied a nearly uniform prior (Cauchy width = 2.0). By the eighth focus group, participant responses provided strong evidence (BF = 11.94) that ATHEMOS did not perform up to the recreational games standard. The average effect size of the difference settled at -0.40 $SD$ (95% CI = -0.61–-0.16), or moderately below the reference data mean (see Table 2). To check the robustness of these results, we tested several alternative priors, as well as a non-directional test, but came to nearly identical conclusions by the eighth focus group (i.e., convergence). We then conducted a similar single-sample t-test with the pilot study data using a fresh prior on $\delta$ (Cauchy width = 2.0), given the uniqueness of this group's experiences with the game. The results of the total score analysis provided very weak evidence that the pilot study impressions were also below that of the recreational games standard (BF = 1.04; $\delta$ = -0.51 [95% CI = -1.00–-0.07]).

Next, we compared the pilot study and focus group results on the GUESS total score to see if there were any between-group differences, with a special focus on whether participant perceptions in the pilot study deviated meaningfully from the eight focus groups. Prior to this analysis, we applied a square root transformation to correct model misspecification (i.e., negative skew). We then performed a Bayesian ANCOVA on the GUESS total score, with group designations entered as fixed factors and biological sex of the participant (i.e., boys versus

**Table 2. Focus Group and Pilot Study Total Score Means Compared to the GUESS Reference Sample Mean (5.71) using Bayesian Single Sample T-Tests.**

|  | F1 | F2 | F3 | F4 | F5 | F6 | F7 | F8 | Pilot |
|---|---|---|---|---|---|---|---|---|---|
| Groupwise |  |  |  |  |  |  |  |  |  |
| $n =$ | 9 | 8 | 7 | 14 | 10 | 10 | 9 | 5 | 16 |
| $M =$ | 5.36 | 5.39 | 4.34 | 5.17 | 5.23 | 5.84 | 5.85 | 4.56 | 5.16 |
| $SD =$ | 0.89 | 0.97 | 1.46 | 1.37 | 0.73 | 0.60 | 0.84 | 0.70 | 1.17 |
| Cumulative |  |  |  |  |  |  |  |  |  |
| $n =$ | 9 | 17 | 24 | 38 | 48 | 58 | 67 | 72 | 16 |
| $M =$ | 5.36 | 5.37 | 5.07 | 5.11 | 5.13 | 5.26 | 5.34 | 5.28 | 5.16 |
| $SD =$ | 0.89 | 0.90 | 1.16 | 1.22 | 1.13 | 1.09 | 1.07 | 1.07 | 1.17 |
| $BF =$ | 0.55 | 0.78 | 6.71 | 18.54 | 52.08 | 8.48 | 3.92 | 11.94 | 1.04 |
| $\delta =$ | -0.48 | -0.43 | -0.60 | -0.55 | -0.57 | -0.41 | -0.34 | -0.40 | -0.51 |
| High CI = | -0.04 | -0.05 | -0.18 | -0.21 | -0.27 | -0.14 | -0.10 | -0.16 | -0.07 |
| Low CI = | -1.14 | -0.90 | -1.03 | -0.90 | -0.87 | -0.67 | -0.59 | -0.65 | -1.00 |

Note. BF = Bayes Factor for one-sample Bayesian t-test; CI = credible interval.

girls) as a covariate. This analysis compared the relative power of four models—group, biological sex, group and biological sex combined, and a null model—when predicting GUESS total score. Results suggest that the data increased the model odds for only biological sex ($BF_M =$ 7.79), which was also the most probable model, P(M|data) = 0.72. After performing Bayesian model averaging, it appeared that the data were 15.63 times more likely for models containing biological sex as a predictor, but only 0.31 times as likely when including group designation. From this, we conclude that only biological sex influenced GUESS total score, and group designation (focus groups or pilot study) had no effect. In other words, participants in the pilot study had impressions of ATHEMOS that were virtually identical to the focus group participants. The results from the satisfaction questionnaire assessing the overall CABI intervention package appeared to confirm these results, with pilot study participants reporting a high degree of satisfaction ($M = 3.34$; $SD = 0.54$; range = 2.13 to 4.00), consistent with responses like *most of my needs have been met* and *mostly satisfied*.

## Discussion

We developed a novel videogame (ATHEMOS) and intervention package for ADHD based on the training interventions used in the *Challenging Horizons Program*. We then examined user acceptability and satisfaction with the game and the broader intervention approach across eight focus groups and a small pilot study during a four-year development project. We found that middle school students with ADHD are engaged by the game, even when it is delivered within a school-based intervention. Although participants did not rate the game as strongly as gamers rate purely recreational games, the differences were only moderate. We believe this is encouraging because educational games can score much lower than recreational games, in the rare instances when this comparison is examined [31]. We noted that players tended to rate ATHEMOS relatively high on the Enjoyment subscale of the GUESS, and relatively low on items measuring Narratives (i.e., the story, characters, and events in the game). One potential explanation is that we designed the game with the expectation that a teacher mentor would help the player understand the game narrative without lengthy in-game exposition, but this might have proved insufficient. It is also possible that players were only moderately engaged by the space invaders storyline, and future modifications might involve distributing the

narrative throughout the game in a more satisfying way, or developing the NPCs to make those interactions more meaningful [32].

Importantly, boys and girls expressed differing opinions of ATHEMOS, with boys appearing more interested than girls. This finding was not surprising, as previous research has shown that boys tend to play videogames at a significantly higher rate than girls, with both sexes playing most often during their middle school years (i.e., ages 11-13) [19]. Both boys and girls enjoy the competitive aspect of videogames, playing against others or their own previous performances, but boys are particularly attuned to the competition provided within "physical" game genres (i.e., fighters, shooters) [20], like ATHEMOS. We conducted a *post-hoc* test examining only the boys' responses to the GUESS ($n$ = 62) and found weak evidence (BF = 0.80) suggesting that their responses approximate the recreational games standard ($\delta$ = -0.23 [95% CI = -0.48–0.01]), but these results were not robust. We conclude from these data that perceptions of ATHEMOS are not dramatically different from purely recreational games. Young adolescents with ADHD, especially boys, appear to be engaged by the game almost as much as they are leisure games. Girls were especially critical of ATHEMOS on the Usability/Playability subscale of the GUESS, which measures how easy it is to learn to play the game. This impression might be attributable to prior experiences with similar videogames (e.g., dual-joystick gamepad); and indeed, the baseline RCGAS data suggested that boys ($M$ = 4.02; $SD$ = 0.82) entered the study moderately more confident than girls ($M$ = 3.55; $SD$ = 0.89) with videogames (BF = 2.86; $\delta$ = 0.50 [95% CI = 0.06–0.96]). Moving forward, we believe that modifications to appeal to girls would improve ATHEMOS, perhaps by offering an alternative minigame to acquire the game's intel or by simplifying the game controls.

Perhaps most encouraging, player perception of ATHEMOS did not meaningfully change when the game was introduced by a teacher interventionist and integrated within a school-based behavior intervention. We were concerned that, once delivered in schools with clear educational goals, players would lose interest. Players who struggle in a videogame can become discouraged, but our results suggest that this did not occur for youth with ADHD playing ATHEMOS. It was also conceivable that players would become critical of the game when asked to play it repeatedly over several weeks, while teachers monitored progress and linked it to real-world goals. But the results suggest that, to the contrary, players found the game just as enjoyable when embedded within a broader CABI approach as they did when encountering the game for the first time in a low-stakes setting.

We believe that positive user responses to ATHEMOS might be partly attributable to the balance between the educational minigames and the recreational elements of the game. The three educational minigames are not time-limited but are likely to consume a small fraction of overall playtime. The space battles are limited to 3 minutes each, but time spent exploring the Silent Canopy base and the in-game upgrades system is unlimited. As a result, players might spend much more time on recreational elements as compared to educational elements. In our view, this imbalance is acceptable because adolescents with ADHD generally have low motivation for intervention. Players who spend disproportionate time on recreational elements can be identified by teacher mentors and then helped to "beat" the minigames, prior to shifting to real-world interventions. We are hopeful that teacher mentors can take advantage of the enjoyable and rewarding nature of ATHEMOS to make this initial game-based feedback pleasant and strengthen the working alliance early in the helping relationship.

## Limitations

There are several limitations to the present study that readers must consider when interpreting our results. First, the GUESS videogame satisfaction scale was validated with adult gamers and has not been extensively tested with young adolescents. Still, we chose this scale because it is

one of the few psychometrically validated satisfaction scales available. The internal reliability estimates of the GUESS subscales in the present study were similar to the reference sample [26], but youth may be more or less severe in their game satisfaction ratings than adults. Second, our results are complicated by the fact that boys and girls clearly responded differently to ATHEMOS. We believe this fits a broader pattern of receptivity to videogames in general [19, 20], but clearly, our overall estimates of acceptability and satisfaction mask these differences. Third, the focus of this study was on playability, acceptability, and satisfaction with ATHE-MOS, which might predict user willingness to play the game and perhaps engage with a teacher mentor, but not necessarily behavior change. More research is needed to examine real-world behavioral outcomes, and whether those changes can be attributed to improvements in the intermediate skills targeted by the game [24].

## Conclusion

In general, we believe this development project shows that games based on psychosocial TIs offer a promising new direction in ADHD games. Research is needed to demonstrate that ATHEMOS leads to meaningful real-world outcomes (e.g., positive outcomes on parent and teacher behavior ratings), but we can conclude from the present study that the target population finds the game concept enjoyable and the intervention approach satisfactory. We believe a game-assisted TI approach offers a research-informed alternative to the cognitive training games that currently dominate this genre of serious games, but that have yet to produce convincing evidence of clinical benefit.

## Author Contributions

**Conceptualization:** Brandon K. Schultz, Steven W. Evans.

**Data curation:** Brandon K. Schultz, Kaitlynn Carter, Emma E. Rogers.

**Formal analysis:** Brandon K. Schultz.

**Funding acquisition:** Brandon K. Schultz, Steven W. Evans.

**Investigation:** Brandon K. Schultz, Steven W. Evans.

**Methodology:** Brandon K. Schultz, Steven W. Evans.

**Project administration:** Brandon K. Schultz, Steven W. Evans, John Bowditch.

**Resources:** John Bowditch.

**Software:** John Bowditch.

**Supervision:** Brandon K. Schultz, Steven W. Evans, John Bowditch.

**Writing – original draft:** Brandon K. Schultz, Kaitlynn Carter, Emma E. Rogers, Jennifer Donelan.

**Writing – review & editing:** Brandon K. Schultz, Steven W. Evans, John Bowditch, Kaitlynn Carter, Emma E. Rogers, Jennifer Donelan, Allison Dembowski.

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
