## [Decision Letter · Decision Letter 0]

17 Apr 2023

PDIG-D-23-00108

Acceptability and playability of an organization training videogame for young adolescents with ADHD: The development of ATHEMOS

PLOS Digital Health

Dear Dr. Schultz,

Thank you for submitting your manuscript to PLOS Digital Health. After careful consideration, we feel that it has merit but does not fully meet PLOS Digital Health's publication criteria as it currently stands. Therefore, we invite you to submit a revised version of the manuscript that addresses the points raised during the review process.

Please submit your revised manuscript within 60 days Jun 16 2023 11:59PM. If you will need more time than this to complete your revisions, please reply to this message or contact the journal office at digitalhealth@plos.org. Please include the following items when submitting your revised manuscript:

We look forward to receiving your revised manuscript.

Kind regards,

Haleh Ayatollahi

Section Editor

PLOS Digital Health

Journal Requirements:

1. We ask that a manuscript source file is provided at Revision. Please upload your manuscript file as a .doc, .docx, .rtf or .tex.

Additional Editor Comments (if provided):

The manuscript is interesting. I appreciate the authors if they re-organize the content of the manuscript. Please remove unnecessary headings in different parts of the manuscript, such as the introduction and methods sections, focus on the main headings, merge the contents and summarize them if possible. All details related to the game development can be moved to the methods section. Please remove unnecessary figures from the discussion section.

Reviewers' comments:

Reviewer's Responses to Questions

**Comments to the Author**

1. Does this manuscript meet PLOS Digital Health’s publication criteria? Is the manuscript technically sound, and do the data support the conclusions? The manuscript must describe methodologically and ethically rigorous research with conclusions that are appropriately drawn based on the data presented.

Reviewer #1: Yes

Reviewer #2: Partly

Reviewer #3: Yes

2. Has the statistical analysis been performed appropriately and rigorously?

Reviewer #1: Yes

Reviewer #2: N/A

Reviewer #3: Yes

3. Have the authors made all data underlying the findings in their manuscript fully available (please refer to the Data Availability Statement at the start of the manuscript PDF file)?

Reviewer #1: Yes

Reviewer #2: No

Reviewer #3: Yes

4. Is the manuscript presented in an intelligible fashion and written in standard English?

Reviewer #1: Yes

Reviewer #2: Yes

Reviewer #3: Yes

5. Review Comments to the Author

Reviewer #1: Dear authors, 

I believe that this is a great topic of research and definitely relevant. Here are some comments to your manuscript: 

1. Introduction. In the sentence "School-based behavior interventions can alleviate some of the academic and social impairments of ADHD, but there are persistent implementation challenges that require interventions to be designed and

tested specifically for those settings." I lack the references to the evidence which point this out. 

I would move the game development and description to the methods section. 

2. Materials and methods. In the participants section, lines 202-215 belongs to the procedure description and not the description of the participants. And I would move the demographics to the start of the results section and include how you analysed this information in the statistical analyses section. 

3. Results. Include all abbreviations used in tables in the Note part of the table. 

4. Discussion. More reference to how your results compare to previous evidence in the field should be mentioned.

Reviewer #2: This reviewer is not a child psychiatrist or paediatrician, and is not an expert in ADHD, but is aware of the harm caused by amphetamine-like medications, and given the reflexive prescription of those medications, any scalable non-pharmacological intervention is of considerable interest.

However, the premise that such a large proportion of children have that condition would seem likely to undermine the validity of an intervention that appears to be trying to attract the interest of children said to have that condition in order to train them to overcome symptoms. The paper does not include outcome data, or a detailed account of how a study to evaluate the intervention might be conducted, including recruitment, inclusion and exclusion criteria, outcome measurement, or a rationale for how it might work.

Reviewer #3: The manuscript is very interesting, describing the development of a serious game (ATHEMOS).

However some major issues emerged after reading the manuscript.

MAJOR ISSUES

The main issue is that the paper does not describe exactly the aims. There is a very long section about the development of the computer program, but no data regarding the perception and efficacy of ATHEMOS in clinical practice. It is reported in the limitations, that “research is needed to demonstrate… real-world outcomes”, but reading the manuscript is not clear, before the real end of itself. The reports of the eight focus group are not reported, as well the suggestion of the pilot sample (except the results of a psychometric scale, GUESS). I suggest the authors to change the title, to be more focused on what the reader should expect from the manuscript, as well in the abstract and in the introduction. 

Abstract: the sentence “implications of these findings” should be removed from the abstract. It is more useful to include what implications are expected, since most of readers read the abstract before the study itself, and this would provide a more appealing interest in the reader.

In the abstract no mention of the Athemos is reported.

Line 18: a reference is needed.

Line 28-34: to the general audience, the CHP could be something not known. A more extensively explanation of the program is needed.

Furthermore, the series of limitations of the CHP should be more contextualized. An explanation of its costs should be provided. The difference between CHP and CHP-C should be more detailed.

Line 187: provide the reference number of the ethical center.

Line 212: the sample size of the pilot study should be reported.

Table 1: explain “biracial”.

Line 243: the authors reported that the serious game was “revised as needed”. However no mention of what changes were made is present. The authors should describe the process of collecting feedback and the change made to the program.

Line 258: the teacher mentor were trained in the used of the program? How? When?

Line 277: a reference is needed regarding the GUESS scale.

MINOR ISSUES

GPA at line 45 is not defined.

Line 210: correct the punctuation.

6. PLOS authors have the option to publish the peer review history of their article (what does this mean?). If published, this will include your full peer review and any attached files.

**Do you want your identity to be public for this peer review?** For information about this choice, including consent withdrawal, please see our Privacy Policy.

Reviewer #1: No

Reviewer #2: No

Reviewer #3: Yes: dr. Tommaso Martino

---

## [Decision Letter · Decision Letter 1]

14 Jun 2023

PDIG-D-23-00108R1

Acceptability and playability of an organization training videogame for young adolescents with ADHD: The development of ATHEMOS

PLOS Digital Health

Dear Dr. Schultz,

Thank you for submitting your manuscript to PLOS Digital Health. After careful consideration, we feel that it has merit but does not fully meet PLOS Digital Health's publication criteria as it currently stands. Therefore, we invite you to submit a revised version of the manuscript that addresses the points raised during the review process.

Please submit your revised manuscript within 30 days Jul 14 2023 11:59PM. If you will need more time than this to complete your revisions, please reply to this message or contact the journal office at digitalhealth@plos.org. Please include the following items when submitting your revised manuscript:

We look forward to receiving your revised manuscript.

Kind regards,

Haleh Ayatollahi

Section Editor

PLOS Digital Health

Journal Requirements:

2. Please send a completed 'Competing Interests' statement, including any COIs declared by your co-authors. If you have no competing interests to declare, please state "The authors have declared that no competing interests exist". Otherwise please declare all competing interests beginning with twhe statement "I have read the journal's policy and the authors of this manuscript have the following competing interests:"

3. Please amend your detailed Financial Disclosure statement. This is published with the article. It must therefore be completed in full sentences and contain the exact wording you wish to be published.

b. If any authors received a salary from any of your funders, please state which authors and which funders.

4. Figures 1-5 contains screenshots. We are not permitted to publish these under our CC-BY 4.0 license; websites are usually intellectual property and are copyrighted.This includes peripheral graphics of the web browser such as the buttons. We ask that you please remove or replace it.

Additional Editor Comments (if provided):

The research was interesting. Please consider the following minor issues in your revision.

1- Please make sure that you have added appropriate keywords after the abstract.

2- In the introduction section, please remove subheadings, integrate them, and present a full introduction section.

3- In the methods section, please start with the “study design”.

4- Table 1 can be moved to the results section.

5- In the methods section, if your research composed of several phases, you can separate them to help the readers to know what happened at each phase, who the participants were, what the research tool was, etc.

6- All details of the game development and Figures can be moved to the results section to make it more informative. If you separate phases of your research in the methods section, the results section can also be reorganized to show the results of each phase separately.

7- In the discussion section, please start with a summary of findings and then support the results with the existing literature and similar studies.

Reviewers' comments:

Reviewer's Responses to Questions

**Comments to the Author**

1. If the authors have adequately addressed your comments raised in a previous round of review and you feel that this manuscript is now acceptable for publication, you may indicate that here to bypass the “Comments to the Author” section, enter your conflict of interest statement in the “Confidential to Editor” section, and submit your "Accept" recommendation.

Reviewer #1: All comments have been addressed

Reviewer #2: All comments have been addressed

Reviewer #3: All comments have been addressed

2. Does this manuscript meet PLOS Digital Health’s publication criteria? Is the manuscript technically sound, and do the data support the conclusions? The manuscript must describe methodologically and ethically rigorous research with conclusions that are appropriately drawn based on the data presented.

Reviewer #1: Yes

Reviewer #2: Partly

Reviewer #3: Yes

3. Has the statistical analysis been performed appropriately and rigorously?

Reviewer #1: Yes

Reviewer #2: I don't know

Reviewer #3: N/A

4. Have the authors made all data underlying the findings in their manuscript fully available (please refer to the Data Availability Statement at the start of the manuscript PDF file)?

Reviewer #1: Yes

Reviewer #2: Yes

Reviewer #3: Yes

5. Is the manuscript presented in an intelligible fashion and written in standard English?

Reviewer #1: Yes

Reviewer #2: Yes

Reviewer #3: Yes

6. Review Comments to the Author

Reviewer #1: Dear Authors, 

Thank you for addressing the comments made. I don't have any additional comments.

Reviewer #2: The authors have addressed my few comments.

Reviewer #3: Authors answered to all my previous comments, and accordingly modified the manuscript.

I have no further issues.

7. PLOS authors have the option to publish the peer review history of their article (what does this mean?). If published, this will include your full peer review and any attached files.

**Do you want your identity to be public for this peer review?** For information about this choice, including consent withdrawal, please see our Privacy Policy. 

Reviewer #1: No

Reviewer #2: No

Reviewer #3: Yes: dr. Tommaso Martino

---

## [Editor Report · Decision Letter 2]

22 Sep 2023

Acceptability and playability of an organization training videogame for young adolescents with ADHD: The development of ATHEMOS

PDIG-D-23-00108R2

Dear Dr. Schultz,

We are pleased to inform you that your manuscript 'Acceptability and playability of an organization training videogame for young adolescents with ADHD: The development of ATHEMOS' has been provisionally accepted for publication in PLOS Digital Health.

Best regards,

Hamish S Fraser, MBCHB MSc

Section Editor

PLOS Digital Health